# Association of Maternal Gestational Vitamin D Supplementation with Respiratory Health of Young Children

**DOI:** 10.3390/nu15102380

**Published:** 2023-05-19

**Authors:** Fanny Loddo, Steve Nauleau, David Lapalus, Sophie Tardieu, Olivier Bernard, Farid Boubred

**Affiliations:** 1APHM, Neonatal Unit, Hospital University la Conception, 13005 Marseille, France; fannylo@orange.fr; 2Regional Health Agency, Provence-Alpes-Côte d’Azur, 13005 Marseille, France; steve.nauleau@ars.sante.fr (S.N.); david.lapalus@ars.sante.fr (D.L.); olivier.bernard@ars.sante.fr (O.B.); 3APHM, Public Health and Medical Information Department, EA 3279, CEReSS–Health Service Research and Quality of Life Centre, 13005 Marseille, France; sophie.tardieu@ap-hm.fr; 4Aix-Marseille Université, C2VN, INRAe, INSERM, 13005 Marseille, France

**Keywords:** vitamin D deficiency, vitamin D supplementation, asthma, bronchiolitis, respiratory tract infections, early term infants

## Abstract

This study aimed to evaluate the association between maternal gestational Vitamin D3 supplementation and early respiratory health in offspring. This was a population-based record-linkage study which used data from the French National Health Database System. Maternal Vitamin D3 supplementation consisted of a single high oral dose of cholecalciferol, (100,000 IU) from the seventh month of pregnancy, according to national guidelines. In total, 125,756 term-born singleton children were included, of which 37% had respiratory illness defined as hospital admission due to respiratory causes or inhalation treatment up to 24 months of age. Infants prenatally exposed to maternal Vitamin D3 supplementation (n = 54,596) were more likely to have a longer gestational age (GA) at birth (GA 36–38 weeks, 22% vs. 20%, *p* < 0.001 in exposed vs. non-exposed infants, respectively). After adjusting for the main risk factors (maternal age, socioeconomic level, mode of delivery, obstetrical and neonatal pathology, birth weight appropriateness, sex, and birth season), the risk of RD was found to be 3% lower than their counterparts (aOR [IC 95%], 0.97 [0.95–0.99], *p* = 0.01). In conclusion, this study provides evidence for the association between maternal gestational Vitamin D3 supplementation and improved early respiratory outcomes in young children.

## 1. Introduction

Pregnant women are a vulnerable population at high risk of vitamin D insufficiency or deficiency. Vitamin D deficiency (blood 25-hydroxyvitamin D [25(OH)D] levels below 50 nmol/L) is common during pregnancy worldwide; in France, the rate was observed to reach 68% [1,2,3]. This is concerning because maternal vitamin D deficiency has been found to be associated with adverse pregnancy outcomes including preeclampsia, glucose intolerance, and low birth weight [4,5,6]. The fetus cannot produce vitamin D and depends exclusively on the mother’s vitamin D status; the maternal-to-fetal transfer occurs mainly during the last trimester of pregnancy [7]. Thus, maternal vitamin D deficiency can in turn impact the development of the fetus. Therefore, many obstetrical societies’ guidelines recommend supplementing pregnant women with Vitamin D3 to support fetal bone metabolism and prevent early neonatal hypocalcemia.

In addition to its role in bone homeostasis and growth, there is evidence that vitamin D plays a biological role in immune function and fetal lung development [8,9]. Animal models of maternal gestational vitamin D deficiency have been reported to impair the bronchiolo–alveolar structure, reduce alveolar density and lung volume in the fetus, and affect long-term pulmonary dysfunctions [8,10]. Observational studies have shown a relationship between maternal gestational vitamin D deficiency and respiratory diseases in preterm and term infants, including respiratory distress syndrome, wheezing events, asthma, and hospital admissions for respiratory tract infections during infancy [11,12,13,14,15,16]. However, randomized studies have reported inconsistent effects relating to the maternal gestational Vitamin D3 supplementation of the offspring’s respiratory outcomes [17,18,19,20].

Although their frequency has decreased worldwide, respiratory diseases remain a frequent cause of hospital admission and mortality in young children, particularly in those younger than 2 [21]. Moreover, respiratory diseases contracted early in life predispose individuals to lung dysfunction, asthma, chronic obstructive disease, and mortality in adulthood [22,23,24]. Further studies are needed to investigate perinatal protective strategies, including maternal Vitamin D3 supplementation that can prevent or mitigate early-life respiratory diseases.

Using the French National Health Database System, we aimed to evaluate the association between maternal Vitamin D3 supplementation during pregnancy and early respiratory outcome in a population of term-born infants living in the Provence–Alpes–Côte d’Azur (PACA) region of France.

## 2. Methods

### 2.1. Study Design and Population

This regional population-based record-linkage study was a retrospective analysis of a prospective cohort that used data from the French National Health Database System (Système National des Données de Santé, SNDS), which includes the National Uniform Hospital Discharge Database (*Programme de Médicalisation du Système d’Information*, (PMSI)) and the National Health Care Insurance Database (*Système National d’Information Inter Régimes de l’Assurance Maladie*, *SNIIRAM*) [25]. The French National Health Database System systematically collects administrative and medical information on each hospitalization incident, including obstetrical and infant characteristics at birth and all healthcare utilizations (drugs, medical and paramedical visits, biological, and radiological examinations). The PMSI database is based on diagnosis-related groups using the International Classification of Diseases, Tenth Revision (ICD-10) (Agence Technique d’Information sur l’ Hospitalization https://www.atih.sante.fr/, accessed on 12 January 2022) [26]. All data were completely anonymized and written informed consent from patients was not required. In France, all regional health agencies have access to this database, in compliance with French law after the French Commission for Data Protection and Liberties (CNIL) deliberation (Décret n° 2016–1871). This study received ethical approval from the French Society of Pediatrics (CERSFP_2022_142).

This study was part of a regional project that aimed to evaluate and improve perinatal health care in the Provence–Alpes–Côte d’Azur (PACA) region of France. The study population included all singleton infants born within the Provence Alps Cote d’Azur (PACA) region between 1 January 2016 and 21 March 2019 with a gestational age (GA) of at least 36 weeks, and for whom data were available for up to 24 months of age. Infants born with congenital anomalies or genetic disorders (Appendix A) who were initially hospitalized or who died during an initial hospital stay, or who required a prolonged initial maternity stay of ≥10 days (as the mean maternity length of stay in France is about 4 to 5 days), were excluded. Infants with no data available after postpartum discharge were also excluded. In the qualitative procedures, cases in which the number of infants was not specified, those with an aberrant weight for GA, and charts without maternal address codes or qualitative issues owing to linking problems (including absence or errors in mother data, mother to infant linkage, coding errors, doubletons) were not included in the analysis (Figure 1). These qualitative procedures were necessary to ensure a cohort with high-quality and complete data.

### 2.2. Primary Outcome

Respiratory illness was defined as at least one hospital admission for respiratory causes or the use of inhaled drugs (bronchodilators, corticosteroids, or combined therapy) for up to 24 months of age. The respiratory causes of hospitalization were determined according to International Classification of Diseases (ICD) codes and classified as follows: viral (ICD-10 codes J10–J12) or bacterial (ICD-10 codes J13–J15) infections, acute bronchitis (ICD-10 codes J20) and bronchiolitis (ICD-10 codes J21), pulmonary infections of unknown origins (ICD-10 codes J18, J22), asthma (ICD-10 codes J45, J46), and bronchiectasis (ICD-10 codes J47). The healthcare insurance system only reimburses pharmacies for the delivery of inhaled medications. Salbutamol (National drugs coding Club Inter-Pharmaceutics [CIP], R03AC02), Fluticasone (CIP, R03BA05), Salmeterol (CIP, R03AC12), Fluticasone combined with Salmeterol (CIP, R03AK06), and Budesonide (CIP, R03BA02) usage data were collected.

### 2.3. Maternal Gestational Vitamin D Supplementation

French guidelines recommend a single high dose of Vitamin D3 (cholecalciferol, 100,000 IU, orally) during the seventh month of pregnancy [3]. This supplementation is exclusively delivered by pharmacies and is universally reimbursed; delivery is thus included in the Health Care Insurance Database. In our study, we defined maternal vitamin D supplementation (cholecalciferol [CIP, A11CC05], 100,000 IU) as drugs delivered by the pharmacies at least 4 weeks before delivery.

It should be noted that French guidelines recommend daily Vitamin D3 supplementation (cholecalciferol 800 IU per day) for children until the age of 2 years.

### 2.4. Maternal and Perinatal Data

Maternal, obstetric, and neonatal data were obtained from maternal and initial mother delivery and infant hospital discharge databases (PMSI). Information on maternal age, obstetric pathology, and mode of delivery (i.e., cesarean section, ICD-10 code O82) was available. Obstetric pathology was determined according to the presence of one of the following complications: preeclampsia (ICD-10 codes O12–O15), gestational diabetes (ICD-10 code O24.4), placental abruption and praevia (ICD-10 codes O44 and O45), preterm premature rupture of membranes (ICD-10 code O42), and chorioamnionitis (ICD-10 code O41.1). Obstetric ultrasound (ECHO) practices defined the quality of obstetric follow-up, according to the French guidelines, as follows: adequate (at least three ECHO at each trimester), intermediate (two ECHO), and inadequate (one or no ECHO) [27].

The GA, sex, and birth weight were recorded. Small-, appropriate-, and large-for-GA (SGA, AGA, and LGA, respectively) were defined as birth weights lower than the 10th percentile, within the 10th and 90th percentiles, and higher than the 90th percentile for sex and GA, based on the national French growth chart [28]. Additionally, data on the birth season were collected (summer–autumn: birth from June to December). Neonatal pathology was defined as the presence of one of the following severe complications: birth with perinatal asphyxia (ICD-10 codes P20 and P21), respiratory distress syndrome (ICD-10 code P22), or meconium aspiration syndrome (ICD-10 code P24). Socioeconomic status was determined at the individual or family level based on healthcare insurance coverage and at the geographical level using the French Deprivation Index (FDep15). Complementary healthcare insurance coverage (CHCI), which provides universal healthcare coverage, is dedicated to the most disadvantaged families based on their home income. FDep15 was calculated using four key socioeconomic and demographic variables extracted from the 2015 census data linked to each local residential municipality (French National Institute of Statistics and Economic Studies (*Institut National de la Statistique et des Etudes Economiques*)): median household income, unemployment rate, percentage of people aged ≥15 years who graduated from high school, and percentage of blue-collar workers in the active population. FDep15 scores ranged from level 1 (affluent) to level 5 (most disadvantaged) [29,30]. This deprivation index is based on the mother-commune postcode. In this study, we compared infants living in the most affluent neighborhoods to those in other areas.

### 2.5. Data Analysis

Univariate parametric and nonparametric analyses were performed using the χ^2^ test for categorical variables, and the Student’s *t*-test or Mann–Whitney U test was performed for quantitative variables, as appropriate. Multivariable regression analysis was performed to estimate the association between maternal gestational vitamin D supplementation and respiratory disease in infants up to 24 months of age. Available variables were included in the final regression model. A two-sided *p*-value < 0.05 was considered statistically significant. Odds ratios (ORs) with 95% confidence intervals (95% CIs) were calculated. Statistical analysis was performed using SAS 9·4 (Satistical Analysis System Institute, Cary, NC, USA).

## 3. Results

Of the 144,697 eligible singleton newborns delivered between 1 January 2016 and 31 March 2019, we enrolled 125,756 infants in our analysis after performing compatibility and reliability measures and after the inclusion criteria were met (Figure 1). During the study period, 54,596 (43%) infants were born to mothers who received Vitamin D3; they were more likely to live in deprived neighborhoods (20% vs. 14%, *p* < 0.0001) than unexposed infants (Appendix A).

A total of 46,142 (37%) infants had respiratory illness, as defined by at least one hospitalization due to respiratory causes or use of inhaled drugs (bronchodilators or corticosteroids). Approximately 8672 (7%) of infants were hospitalized (79% of them during the first 12 months after birth), and 42,761 (34%) were prescribed inhaled drugs. Infants with respiratory disease were more frequently boys (66% vs. 47%, *p* < 0.0001), LGA at birth (+10%, *p* < 0.001), and born with a shorter GA (GA less than 38 weeks, (n = 10,679/46,142) 23% vs. (n = 16,341/79,614) 20% *p* < 0.0001) (Table 1). Based on these results, it was estimated that 200 pregnant women need to be supplemented to prevent one child with respiratory disease.

After a multivariable regression logistic analysis, infants prenatally exposed to maternal Vitamin D3 had 3% lower odds of having a respiratory disease than unexposed infants, independent of maternal age, socioeconomic status, mode of delivery, birth weight appropriateness, sex, obstetrical and early neonatal complications, and birth season with an adjusted OR (95% CI) of (aOR = 0.97 (0.95–0.99), *p* = 0.01) (Table 2). Furthermore, the analysis revealed that a shorter GA was associated with respiratory disease (GA 36 weeks vs. 39 weeks, aOR = 1.26 [1.16–1.38], *p* < 0.0001), while female sex (aOR = 0.70 [0.68–0.71], *p* < 0.0001) and aged mother (≥40 years vs. 20–29 years, aOR = 0.85 (0.80–0.90), *p* < 0.001) were protective factors.

## 4. Discussion

In this large regional population-based data-linkage study of singleton “relatively healthy” term-born infants, we found that children prenatally exposed to a maternal high single dose of Vitamin D3 (cholecalciferol) supplementation in the seventh month of pregnancy had a lower risk of respiratory illness for up to 24 months of age than unexposed infants. This slight reduction was independent of other known perinatal risk factors.

The association between maternal gestational Vitamin D supplementation and children’s respiratory outcomes suggests that vitamin D plays a biological role in lung development. Vitamin D receptor (VDR), a nuclear receptor specifically for the bioactive form of vitamin D (1,25(OH)_2_D3), is expressed in fetal lung cells. A subset of more than 500 genes involved in lung development in humans and rodents contains VDR response elements [31,32]. In preclinical studies, maternal gestational vitamin D deficiency affected pneumocyte type II differentiation, decreased alveolar density, induced airway smooth muscle hypertrophy, and decreased tracheal size [8,10,33]. In the long term, these abnormalities increase airway resistance and reduce lung vital capacity. In humans, two meta-analyses reported an inverse relationship between cord blood 25(OH)D levels and wheezing during infancy (aOR [95% CI] 0.43 [0.29–0.62]; *p* < 0.001) and elevated respiratory tract infections in children born to vitamin D-deficient mothers (aOR [95% CI] of 0.64 [0.47–0.87]) [12,34]. These associations may also be supported by the biological role of vitamin D in the immune system [9,35]. VDR elements are expressed in circulating monocytes, macrophages, dendritic cells, and activated T lymphocytes [36]. Vitamin D supports the immune defense system and limits inflammatory reactions by decreasing lymphocyte proliferation and the production of certain proinflammatory cytokines and by increasing anti-inflammatory cytokine (IL10) [37].

Although there is a relationship between maternal or fetal vitamin D status and children’s respiratory outcomes, the preventive effects of maternal gestational Vitamin D3 supplementation is controversial. In rodents, perinatal 1,25(OH)_2_D administration prevents lung injury associated with chorioamnionitis and restores lung structure in pups prenatally exposed to maternal gestational vitamin D depletion [38,39]. A meta-analyses of two large randomized studies (Vitamin D Antenatal Asthma Reduction Trial (VDAART) and Copenhagen Prospective Studies on Asthma in Childhood (COPSAC) studies), which included 1387 infants, reported that daily Vitamin D3 supplementation in pregnant women from the second trimester until the end of pregnancy (2400 IU and 4000 IU daily, respectively, vs. 400 IU daily) resulted in a 25% reduction in the risk of asthma or recurrent wheezing in three-year-old children (aOR) = 0.74 (95% CI, 0.57–0.96); this effect was particularly relevant in children born to mothers with a sufficient vitamin D status (25(OH)D levels ≥ 30 ng/mL) at randomization [40]. This finding may indicate that (i) fetal vitamin D status early during pregnancy is important for lung development or that (ii) supplementation could be insufficient in severely deficient mothers.

Our findings provide additional evidence, although the dose of maternal Vitamin D3 supplementation was lower and the administration was later during pregnancy. A single maternal high dose of Vitamin D3 (100,000 IU Vitamin D3) at the beginning of the third trimester (between GA 28 weeks and 32 weeks) is equivalent to approximately 1100 IU daily for 3 months. The pulmonary effects observed in our study may result from the pharmacokinetics profile and the timing of the supplementation protocol. After a single high dose of Vitamin D3, blood 25(OH)D levels significantly increased from day 2 and reached a steady state from days 15 to 21 during one month before declining [41]. This rapid rise in 25(OH)D levels in both the mother and the fetus occurs during the formation of the small bronchio–alveolar structures. In the VDAARTrial, low maternal 25(OH)D levels during the third trimester were associated with small airway resistance in four-to-six-year-old children [42]. This supplementation protocol may, however, be insufficient for preventing or correcting maternal vitamin D deficiency, particularly as the proportion of disadvantaged pregnant women, known to be a high-risk population for vitamin D deficiency, was consistently higher in the supplementation group [3]. Although the reduction in respiratory disease after maternal Vitamin D3 supplementation observed in our study is small, with the number of pregnant women requiring treatment being 200, it can be considered relevant if the whole population is considered. All these findings suggest that maternal gestational Vitamin D3 supplementation is associated with a low risk of respiratory disease in young children; however, further studies are needed to better define the optimal protocol.

### Strengths and Limitations

The National Health Database System does not include data on relevant individual factors, including ethnicity, breastfeeding, home environment (parental smoking, air pollution), and allergy antecedents, which could influence infant respiratory outcomes. However, some of these factors are linked to socioeconomic status, which was included in our analysis [43]. Maternal body mass index and obesity/overweight data are not included in the database; however, the association between maternal gestational vitamin D and respiratory outcomes in children remained significant after adjusting for large for gestational age, which is frequently related to maternal overweight/obesity or diabetes. Maternal vitamin D status was also unknown; therefore, we do not know whether it can mediate the effects of maternal vitamin D supplementation. Moreover, the healthcare insurance system does not cover the individual consumption of food enriched in vitamin D (medicinal-like products), which usually contains a low dose of Vitamin D; they are not recommended for pregnant women in France and their use are limited [44]. We do not know how such nutritional practices could influence our findings, but they can be considered as marginal. Finally, our study was not designed to evaluate the effect of postnatal vitamin D supplementation in children. In France, it is recommended for children to take daily Vitamin D3 (cholecalciferol) supplements for up to 2 years of age.

The main strength of this study is that it was based on the National Health Database System, which included data from all public and private hospitals, as well as on the consumption of medical drugs that are exclusively delivered by pharmacies in France. This database system provides robust and high-quality data, making it possible to link prenatal and postnatal pathologies with healthcare utilization in a large population. We used robust criteria to define respiratory disease and assess the use of an inhaled treatment that is considered to be a reliable marker of childhood wheezing pathology. Finally, the use of this system can be an alternative to observational clinical studies that are often difficult to conduct over a long term and to highlight associations in large populations.

## 5. Conclusions

In this large French regional population-based record-linkage study, using the French National Health Database System, we found an association between high single-dose maternal vitamin D supplementation during the seventh month of pregnancy and lower odds of respiratory illness in children, as defined by hospital admission for respiratory reasons or the use of inhaled treatment in a population of term-born infants up to 2 years of age. Our study provides further insight into the biological role of vitamin D in lung and immune system development. Maternal vitamin D supplementation during pregnancy could be a strategy used to prevent early respiratory diseases in young children and reduce adverse pulmonary consequences in the long term. Given the study design, the study population, and limited data, particularly regarding maternal vitamin D status and dietary intake, the generalizability of our results may be questioned. Further studies should include maternal gestational vitamin D status, diet, and lifestyle investigations (e.g., exposure to the sun)**.** Moreover, they should better define the optimal protocol, including the doses and the timing of supplementation.

## Figures and Tables

**Figure 1 nutrients-15-02380-f001:**
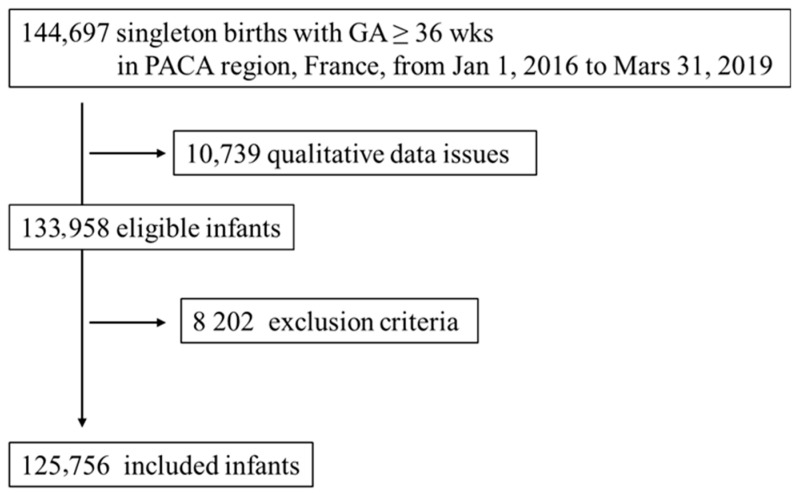
Flow chart of study population.

**Table 1 nutrients-15-02380-t001:** Characteristics of study population.

Characteristics	Overall (n = 125,756)	Respiratory Illness (n = 46,142 [37%])	No Respiratory Illness (n = 79,614 [63%])
Maternal age (years)			
<20	1612 (1%)	550 (1%)	1062 (1%)
20–29	53,920 (43%)	19,914 (43%)	34,006 (43%)
30–39	64,709 (51%)	23,834 (52%)	40,875 (51%)
>40	5515 (4%)	1844 (4%)	3671 (5%)
Pregnancy follow-up			
Inadequate	9319 (7%)	2910 (6%)	6409 (8%)
Intermediate	20,824 (17%)	7251 (16%)	13,573 (17%)
Adequate	95,613 (76%)	35,981 (75%)	59,632 (75%)
CHCI	30,524 (24%)	11,217 (24%)	19,307 (24%)
NDI			
Deprived	20,751 (16%)	7222 (16%)	13,529 (17%)
Medium	75,764 (60%)	28,161 (61%)	47,603 (60%)
Affluant	28,828 (23%)	10,609 (23%)	18,219 (23%)
Obstetrical pathology	19,766 (16%)	7314 (16%)	12,452 (15%)
Maternal Vitamin D3	54,696 (43%)	19,938 (43%)	34,758 (44%)
Caesarean section	15,858 (13%)	5961 (13%)	9897 (12%)
Birth season			
Summer–Autumn	50,999 (40%)	19,130 (41%)	31,869 (40%)
GA (weeks)			
36	2258 (2%)	939 (2%)	1319 (2%)
37	6293 (5%)	2500 (5%)	3793 (5%)
38	18,469 (15%)	7240 (16%)	11,229 (14%)
39	37,689 (30%)	13,999 (30%)	23,690 (30%)
>40	61,047 (48%)	21,464 (46%)	39,583 (50%)
Female sex	62,012 (49%)	20,118 (44%)	41,894 (53%)
Birth weight			
AGA	102,274 (82%)	37,675 (82%)	65,049 (82%)
SGA	11,479 (9%)	4104 (9%)	7375 (9%)
LGA	11,553 (9%)	4363 (10%)	7190 (9%)
Neonatal pathology	15,217 (12%)	5607 (12%)	9610 (12%)

Data are n (%). AGA: birth weight adapted for gestational age, SGA: small for gestational age, LGA: large for gestational age; CHCI: complementary healthcare insurance coverage; NDI: neighborhood deprived index. Relationships for all characteristics and status of respiratory disease were significant (*p* < 0.0001) for all except for CHCI, Obstetrical pathology, Birth weight and Neonatal pathology (*p* < 0.001) and for maternal VitD, for which *p* = 0.001.

**Table 2 nutrients-15-02380-t002:** Maternal gestational vitamin D supplementation associated with lessen respiratory illness of children. Multivariable analysis.

Characteristics	Adjusted OR (95%CI)	*p*
Maternal age (years)		
<20	0.931 (0.84–1.03)	0.18
20–29	Reference [1]	1
30–39	0.98 (0.96–1.01)	0.21
>40	0.84 (0.79–0.89)	<0.001
Pregnancy follow-up		
Inadequate	Reference [1]	
Intermediate	1.12 (1.05–1.19)	<0.0001
Adequate	1.32 (1.25–1.40)	<0.0001
CHCI	1.02 (0.99–1.05)	0.12
NDI (Affluent)	1.00 (0.97–1.03)	0.98
Obstetrical pathology	0.99 (0.96–1.30)	0.85
Maternal Vitamin D3 supplementation	0.97 (0.95–0.99)	0.01
Caesarean section	0.98 (0.95–1.02)	0.38
Birth season		
Summer–Autumn	1.06 (1.03–1.08)	<0.001
GA (weeks)		
36	1.26 (1.16–1.38)	<0.0001
37	1.16 (1.10–1.23)	<0.0001
38	1.11 (1.07–1.15)	<0.0001
39	Reference [1]	1
>40	0.85 (0.80–0.90)	<0.0001
Female sex	0.70 (0.68–0.71)	<0.0001
Birth weight		
AGA	Reference [1]	
SGA	0.97 (0.93–1.01)	0.13
LGA	1.05 (1.01–1.09)	0.01
Neonatal pathology	1.02 (0.99–1.06)	0.16

Data are n (%). AGA: birth weight adapted for gestational age, SGA: small for gestational age, LGA: large for gestational age; CHCI: complementary healthcare insurance coverage; NDI: neighborhood deprived index.

## Data Availability

The French National Health Database System (Système National des Données de Santé, SNDS) can only be accessed through a secure server after obtaining requisite ethical and data protection authorizations. The data are available upon request from the authors.

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
