# Peer review of "Association of Maternal Gestational Vitamin D Supplementation with Respiratory Health of Young Children"

_nutrients, 2023, doi:10.3390/nu15102380_

Round 1

Reviewer 1 Report

Authors present an analysis the association between maternal Vit D supplementation during pregnancy and early respiratory outcome in a population of term-born infants living in the Provence-Alpes-Côte d'Azur (PACA) region of France using the French National Health Database System.

Overall, the study is well presented and interesting, although the patient population is limited to one region in France. It would be necessary to explain why? Is it representative region for the whole France?

Please find below the areas for minor improvement:

Figure 1. Flow chart ... should rather be in the methodology (Study design and population)

Table 1 - Parameters? Why are they not listed in the p-value table?

Considering that these data are from one region of France and that there are many limitations of the study, especially with regard to maternal Vit D status and dietary intake, the results / conclusions of these studies cannot be applied to the general population. Still, the results may begin further research on this topic, including a diet analysis and lifestyle (exposure to the sun, stimulants) and nutritional status etc.

In addition, the manuscript should be adapted to the editorial guidelines of the journal, especially with regard to citing references and unifying the spelling of individual names of ingredients, diseases as well as providing (showing) data from tables in the text.

English style and syntaxis should be revised in depth by a native English speaker.

Author Response

Reviewer 1

We would like to thank you for your comments which have improved the manuscript. Changes appeared in bold in the text.

- Point 1. Overall, the study is well presented and interesting, although the patient population is limited to one region in France. It would be necessary to explain why? Is it representative region for the whole France?

Response: This study was part of a regional project aimed at evaluate and improving perinatal health care in the Provence-Alpes-Côte d'Azur (PACA) region of France and was conducted by the Regional Health Agency which has access to the data from the PACA region only. We agree that we cannot consider this region as representative for the whole France. For example, the PACA region which is located in the south of France, is more exposed to sun than the northern region of France. (Please find the Response to Point 4 below)

- Point 2. Figure 1. Flow chart ... should rather be in the methodology (Study design and population)

Response: Well noticed.

- Point 3. Table 1 - Parameters? Why are they not listed in the p-value table?

Response: “Parameters” has been replaced by “Characteristics”. Because of the large sample size, all of the characteristics were significant. P values are listed in the Table comments.

- Point 4. Considering that these data are from one region of France and that there are many limitations of the study, especially with regard to maternal Vit D status and dietary intake, the results / conclusions of these studies cannot be applied to the general population. Still, the results may begin further research on this topic, including a diet analysis and lifestyle (exposure to the sun, stimulants) and nutritional status etc.

Response: Thank you for your comments. We added the following sentence in the Conclusion section: “Given the study design, study population, and limited data, particularly regarding maternal vitamin D status and dietary intake, the generalizability of our results may be questioned. Further studies should include maternal gestational diet analysis, lifestyle (exposure to the sun, stimulants,...) and nutritional status.”

- Point 5. In addition, the manuscript should be adapted to the editorial guidelines of the journal, especially with regard to citing references and unifying the spelling of individual names of ingredients, diseases as well as providing (showing) data from tables in the text.

Response: changes have been made.

Reviewer 2 Report

This is a very interesting and meaningful study, and I hope that my following suggestions will help to improve the manuscript

(1)    Line 26. Vitamin d deficiency and vitamin D supplementation, Which D please?

(2)    The resolution of Figure 1 needs to be improved.

(3)    The format of references should be uniform.

(4)    Line 35-36. “In animals, maternal gestational Vit D deficiency reduces alveolar density and lung volume in the fetus and has long term effects on offspring pulmonary functions (1,7).” This sentence should be deleted, thus, it’s better to link context.

(5)    Line 160-163. “After a multivariable regression logistic analysis, infants prenatally exposed to maternal Vit D had 3 % lower odds of having respiratory disease than unexposed infants….” Very interesting results, what do you think of this 3% difference in the health of the infant

(6)    Line 226-245. Your points about these limitations are excellent, please could you give the corresponding countermeasure suggestions

Author Response

Reviewer 2

We would like to thank you for your comments which have improved the manuscript. Changes appeared in bold in the text.

-Point 1. Line 26. Vitamin d deficiency and vitamin D supplementation, Which D please?

Response: cholecalciferol

-Point 2. The resolution of Figure 1 needs to be improved.

Response: Changes have been done. Thank you

- Point 3. The format of references should be uniform.

Response: changes have been done.

Point 4. Line 35-36. “In animals, maternal gestational Vit D deficiency reduces alveolar density and lung volume in the fetus and has long term effects on offspring pulmonary functions (1,7).” This sentence should be deleted, thus, it’s better to link context.

Response: Thank you for your comment. We understand that deleting this sentence should improve the flow of the text. Nevertheless, we considered experimental findings relevant for introduction of the study. We propose to move and add some changes to the sentence: “These findings confirmed experimental data. Animal models of maternal gestational vitamin D deficiency reported reduced alveolar density and lung volume in the fetus and long term pulmonary dyfunctions” (Introduction, p.1)

Point 5. Line 160-163. “After a multivariable regression logistic analysis, infants prenatally exposed to maternal Vit D had 3 % lower odds of having respiratory disease than unexposed infants….” Very interesting results, what do you think of this 3% difference in the health of the infant.

Response: this reduction is quite surprising given the limited data available on this topic, the study design (population-based study with inherent biases (population of relatively healthy newborn, unknown maternal vitamin D status, diet and life style,...) and the respiratory disease criteria (Hospitalization for respiratory disease or use of inhaled treatment which could be considered as a marker of wheezing disease). “Although the reduction in respiratory disease after maternal vitamin D supplementation observed in our study is small with the number needed to treat of 200, it can be considered relevant if the whole population is considered (Discussion section, p7).”

Point 6. Line 226-245. Your points about these limitations are excellent, please could you give the corresponding countermeasure suggestions

Response: the National Health Database System cannot provide sufficient countermeasure. “Further randomized or observational studies should include maternal gestational vitamin D status, diet and life style investigations (Discussion, p.7).”
